# CoBIT: A Contrastive Bi-directional Image-Text Generation Model

**Haoxuan You[1][†], Mandy Guo[2], Zhecan Wang[1], Kai-Wei Chang[3], Jason Baldridge[2], Jiahui Yu[2]**
[1]Columbia University, [2]Google Research, [3]UCLA
`haoxuan.you@cs.columbia.edu, {xyguo,jasonbaldridge,jiahuiyu}@google.com`

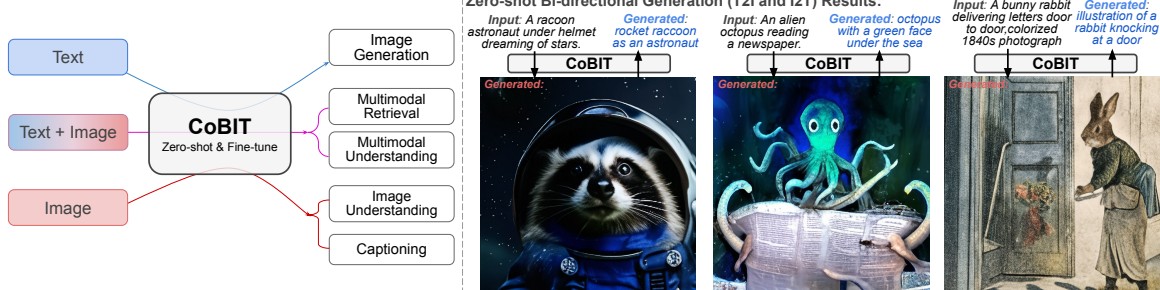

Figure 1: CoBIT can address a variety of vision and vision-language tasks in zero-shot and fine-tuning settings. The right-side displays the zero-shot generated images by CoBIT given novel prompts, and the zero-shot generated captions by CoBIT given the previously generated images as input.

## Abstract

The field of Vision-and-Language (VL) has witnessed a proliferation of pretrained foundation models. Current techniques typically employ only one type of training objective, whether it's (1) contrastive objectives (like CLIP), (2) image-to-text generative objectives (like PaLI), or (3) text-to-image generative objectives (like Parti). However, all these three objectives are mutually relevant and are all based on image-text pairs. Intuitively, the first two objectives can be considered as complementary projections between two modalities, and contrastive learning can preserve global alignment and generations facilitate fine-grained understanding. Inspired by this, we present a Contrastive Bi-directional Image-Text generation model (CoBIT) to first time unify the three pre-training objectives in one framework. Specifically, CoBIT employs a novel unicoder-decoder structure consisting of an image unicoder, a text unicoder, and a cross-modal decoder. The image/text unicoders can switch between encoding and decoding in different tasks, enabling flexibility and shared knowledge that benefits both image-to-text and text-to-image generations. CoBIT achieves superior performance in image understanding, image-text understanding (Retrieval, Captioning, VQA, SNLI-VE), and text-based content creation, particularly in zero-shot scenarios.

## 1 Introduction

Recently, there has been rising interest in developing multimodal foundation models for vision-language tasks. By mapping text and image representation in the same space, the models can (1) generate images from text (Ramesh et al., 2021; Yu et al., 2022b; Chang et al., 2022; 2023), (2) generate captions from images (Wang et al., 2022a; Chen et al., 2022; Wang et al., 2021; Alayrac et al., 2022), and (3) retrieve images from text and vice verse (Radford et al., 2021; Yao et al., 2021; Mu et al., 2022; You et al., 2022). Although these tasks are highly relevant and can be operationalized on the same set of image-text pairs. They are often considered separately, and the corresponding foundation models are trained with different pre-training losses designed for the corresponding task.

---

[†] This work was done when Haoxuan was an intern at Google.

Specifically, there are three pre-training losses that are widely used in the literature: (1) contrastive objectives, (2) image-to-text generative objectives, and (3) text-to-image generative objectives. Most models are trained with only one of these objectives, while some are trained with two. For example, CoCa (Yu et al., 2022a) combines contrastive learning and image-to-text generation. OFA (Wang et al., 2022b) and UnifiedIO (Lu et al., 2022) integret image-to-text and text-to-image generation. However, none of the approaches has considered using all these three losses although they are highly relevant and can be trained with the same set of image-text pairs.

Intuitively, these pre-training objectives complement each other. Specifically, contrastive learning drives high-level image-text matching, whereas image/text generation encourages the model to learn fine-grained image and text representations. Therefore, it is intuitive to utilize them in the same framework. It is worth noting that these three pre-training losses can share part of the computational graphs. Therefore, optimizing them jointly does not increase much overhead compared to only optimizing one.

In this paper, we propose to unify the three commonly used pre-training VL objectives: cross-modal contrastive learning, image-to-text generation, and text-to-image generation, and consolidate their strengths in one framework. Our key innovation is a simple and unified **Co**ntrastive **B**i-directional **I**mage-**T**ext generation model (**CoBIT**), which consists of an *image unicoder* and a *text unicoder*, as well as a cross-attention decoder. The proposed image/text unicoder uses the Transformer architecture. It alternates between two modes: unimodal image/text encoding and decoding depending on the pre-training tasks. Importantly, the same set of Transformer parameters are used for both encoding and decoding, with only the input embedding and attention masks differing. As shown in Fig. 2, when optimizing contrastive objective, image unicoder, and text unicoder work as two encoders. When optimizing text/image generation loss, image/text unicoder extracts features in encoding mode, and the text/image unicoder works in autoregressive decoding mode, then the cross-attention decoder will let autoregressive text/image features cross-attend to encoded image/text feature, serving as a fuser and generator. Each unicoder efficiently shares the knowledge between encoding and decoding and, therefore, can jointly improve both T2I and I2T generation without increasing the number of parameters, exhibiting excellent parameter efficiency. In such a way, all three pre-training paradigms are unified in our framework.

Our extensive experiments demonstrate CoBIT's superior performance, and more importantly, **first time** verifies the compatibility of the three objectives. Benefiting from the compatible objectives, CoBIT subsumes strong zero-shot and transferable capacities of unimodal visual understanding, image-text matching, image-text understanding, and text-to-image generation. For example, CoBIT achieves 82.7% accuracy in zero-shot ImageNet classification, 9.37 FID in zero-shot text-to-image generation, 44.8 CIDEr score in zero-shot image-to-text captioning. After fine-tuning, CoBIT further achieves 86.44% linear probing accuracy on ImageNet, 4.62 FID on text-to-image generation, and 78.3 VQA score.

## 2 RELATED WORK

**Learning Visual Representation from Text.** Recent works studied pre-training a visual backbone supervised by paired text data and produced transferable visual representations. CLIP (Radford et al., 2021) and ALIGN (Jia et al., 2021) are prominent examples of global contrasting between image-text pairs. Florence (Yuan et al., 2021), BASIC (Pham et al., 2021), and LiT (Zhai et al., 2022b) further scale both datasets and models. FILIP (Yao et al., 2021) proposes to employ local token features from images and text for fine-grained contrastive learning. MS-CLIP (You et al., 2022) and CLIPPO (Tschannen et al., 2022) study sharing the model parameters between vision and text.

**VL Pre-training.** Another line of research focuses on learning a solid joint multimodal embedding through pre-training. Some pre-train with mask-reconstruction loss (Li et al., 2019; Wang et al., 2022c; Li et al., 2022; Chen et al., 2019; Shen et al., 2021; Li et al., 2021), *i.e.*, mask partial image and text tokens in input and require the model to predict the masked tokens. Others pre-train models by generating text autoregressively (Wang et al., 2021; Chen et al., 2022; Wang et al., 2022a; Alayrac et al., 2022). Both perform strongly in downstream VL understanding tasks, such as VQA (Antol et al., 2015) and captioning.

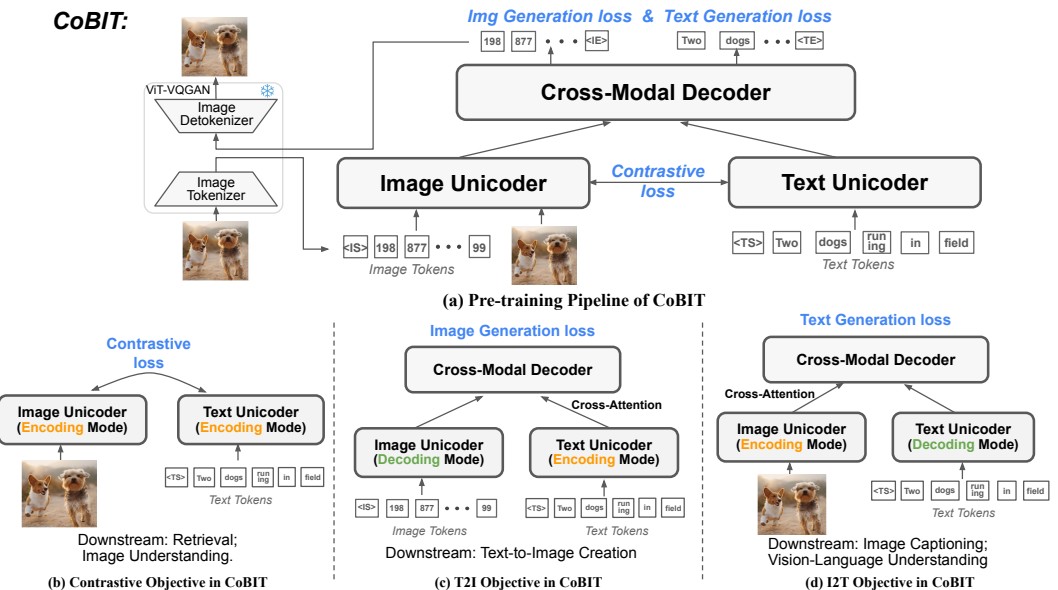

Figure 2: (a): Overview of CoBIT pre-training pipeline; (b): When optimizing contrastive objective, image unicoder and text unicoder work as two encoders; (c) and (d): When optimizing image/text generation loss, text/image unicoder extracts features in encoding mode and image/text unicoder works in autoregressive decoding mode, then the cross-attention decoder will let autoregressive image/text features cross-attend to encoded text/image feature.

**Text-to-Image Generation.** Text-guided image creation is a challenging problem that has attracted intense interest in the past two years. Two families of methods are widely studied: diffusion-based and token-based. Diffusion-based models (Rombach et al., 2022; Saharia et al., 2022; Ramesh et al., 2022) are based on a process that iteratively adds noise to images and then learns to reverse the noising process while conditioning on textual descriptions of the image. With token-based methods, raw images are quantized into image tokens by an image tokenizer; then, given a text input, Transformer models predict image tokens autoregressively like machine translation (Ramesh et al., 2021; Yu et al., 2022b) or by iteratively predicting image tokens in parallel(Chang et al., 2022; 2023).

As these three broad lines of research have demonstrated great transferable ability to various downstream tasks, there have been many efforts to unify some of them (Yu et al., 2022a; Wang et al., 2022b; Lu et al., 2022; Zhang et al., 2021a; Kim et al., 2022; Huang et al., 2021). Our work, CoBIT, serves as the first effort to integrate contrastive loss, image-to-text generation, and text-to-image loss under one unified pre-training framework.

## 3 CoBIT

We begin with describing the input processing and then present the model architecture, which includes a proposed unicoder module that shares the merit of both unimodal encoding and decoding. Finally, we explain the pre-training of CoBIT and discuss a comparison with other unified models.

### 3.1 INPUT

To cover various tasks, CoBIT supports three inputs: text tokens, discrete image tokens, and raw images.

**Text Tokens.** Following the default process in past works (Raffel et al., 2020; Jia et al., 2021; Yu et al., 2022a), we tokenize text inputs using a SentencePiece model with a 64k vocabulary trained on the sampled pre-training datasets. The maximum text token length is 64.

**Discrete Image Tokens.** CoBIT generates images in an autoregressive manner, which requires tokenizing 2D images into a sequence of image tokens (Ramesh et al., 2021; Ding et al., 2021; 2022; Gafni et al., 2022; Yu et al., 2022b). Following Parti (Yu et al., 2022b), we employ a pre-trained and frozen ViT-VQGAN (Yu et al., 2021) as the tokenizer. Specifically, each 256×256 image is tokenized into a 32×32 grid of image tokens, with 8192 image token classes in the codebook.

| Model | Image Unicoder | | Text Unicoder | | Cross-modal Decoder | | Total Params |
|---|---|---|---|---|---|---|---|
| | Layers | Dims | Layers | Dims | Layers | Dims | |
| CoBIT-Base | 12 | 768 | 12 | 768 | 18 | 1024 | 626M |
| CoBIT-Large | 20 | 1024 | 12 | 1024 | 30 | 1024 | 1082M |

Table 1: Size variants of CoBIT.

We append the codebook to the text vocabulary as additional tokens. In inference, to generate images, we decode the image tokens one-by-one and feed them into the decoder in ViT-VQGAN to reconstruct the raw image.

**Raw Image.** For image and image-text understanding tasks, we input raw images, and each image is divided into non-overlapped patches following the de facto process in ViTs. In default, unless specified, the image resolution is 288x288, and the patch size is 18x18.

## 3.2 ARCHITECTURE

As shown in Fig. 2, CoBIT comprises one image unicoder, one text unicoder, and one cross-attention decoder. We term them unicoders because they can act as either encoders or decoders, depending on the role they play for each task. The incorporation of text/image unicoder is inspired by Dong et al. (2019); Bao et al. (2020); Zhou et al. (2020), which demonstrated that one Transformer model can perform both bidirectional encoding for understanding tasks and autoregressive decoding for generation tasks. In our scenario, compared with plain image/text encoders, unicoders in decoding mode can take advantage of the common knowledge shared with encoding to produce unimodal autoregressive features as a decent prior for cross-modal generative objective. Experimental ablation also validates that unicoders boost both T2I generation and multimodal understanding.

**Image Unicoder.** Recently, Vision Transformers (ViT) (Dosovitskiy et al., 2020; Touvron et al., 2021; Liu et al., 2021) has been established as the strongest approach for image feature *encoding*. As *decoders*, Transformers are used in autoregressive image token generation (Ramesh et al., 2021; Gafni et al., 2022; Yu et al., 2022b). We combine these two functionalities into a single image unicoder. The image unicoder has two working modes: (1) In the encoding mode, following ViT, each 2D patch in the raw image is projected into a feature vector by a trainable linear projection layer. Then, the projected feature sequence is input into cascaded Transformer layers to obtain the encoded image features, where the attention mask is bi-directional. (2) In the decoding mode, firstly, the input processing is different. As described in Sec. 3.1, we tokenize the raw image into image tokens and initialize an embedding layer where token embeddings are indexed. Then, the same Transformer layers in encoding mode are reused in decoding mode to process the features; however, to guarantee the causal decoding ability, we use causal conv-shaped attention mask (Ramesh et al., 2021; Yu et al., 2022b; Child et al., 2019) instead. Overall, the two modes share the Transformer layers' parameters, and only differ in input processing and attention masks. We assume that, compared with the design of plain image encoders as in previous works (Yu et al., 2022a; Wang et al., 2022a), the additional decoding mode can exploit the common knowledge learned in image encoding to generate image autoregressive features, which we hypothesize should boost the (text-to-)image generation capacity.

**Text Unicoder.** Similar to the image unicoder mentioned above, the text unicoder also has both encoding and decoding modes, which reuse the Transformer parameters. In both modes, the same tokenizer and embedding layer are utilized to obtain token features, given that they share the same input formats. A causal attention mask is applied in decoding mode. During encoding of text, there are two options in previous works: bi-directional mask (Devlin et al., 2018; Raffel et al., 2020; Yu et al., 2022b), or causal mask (Brown et al., 2020; Radford et al., 2021; Yao et al., 2021). We empirically found that two masks make no difference in performance and use causal masking as the default in the reported experiments.

**Cross-modal Decoder** The cross-modal decoder performs as a fusion-and-generation module, which structure-wise follows the cross-attention decoder (Vaswani et al., 2017; Yu et al., 2022a). When generating text, the input is the text autoregressive feature from the text unicoder in decoding mode; encoded image features will be treated as cross-attention information, *i.e.*, key and value in cross-attention layers. When generating the image, symmetrically, the image token autoregressive feature from the image unicoder in decoding mode is input and cross-attends to encoded text features. Also, different from text generation, where the plain causal (autoregressive) mask is used in the cross-modal decoder, image generation employs a conv-shaped masked sparse attention (Ramesh

et al., 2021; Yu et al., 2022b; Child et al., 2019), which can save memory and computation brought by long sequences of image tokens.

## 3.3 PRE-TRAINING

The pre-training of CoBIT subsumes three fundamental objectives: image-text contrastive loss, I2T generation loss and T2I generation loss. Here, we provide details on the losses and also clarify the scaling and initialization strategy.

**Contrastive Loss.** We input raw image and text into the image unicoder and the text unicoder, respectively (both in encoding mode), to get encoded image and text features. For text, as with CLIP (Radford et al., 2019) and ALIGN (Jia et al., 2021), we take the feature vector of the CLS token appended at the end of the input sequence as the global representation. For images, however, the unicoder outputs a sequence of features. To aggregate them, following (Yu et al., 2022a), we apply an attention pooler, which is a single multi-head attention layer with one learnable query and unicoder output features as key and value. After obtaining two global features of image and text, a contrastive loss is applied to optimize the paired image-text against others in the same batch:

$$\mathcal{L}_{\text{Con}} = -\frac{1}{N}(\sum_i^N \log \frac{\exp(x_i^T y_i/\tau)}{\sum_{j=1}^N \exp(x_i^T y_j/\tau)} + \sum_i^N \log \frac{\exp(y_i^T x_i/\tau)}{\sum_{j=1}^N \exp(y_i^T x_j/\tau)}), \tag{1}$$

where $x_i$ and $y_j$ denote the normalized global embeddings of $i$-th image and $j$-th text. $\tau$ is a learnable temperature for adjusting the scale of the loss.

**I2T and T2I Generation Loss.** We formulate two generation tasks as token generation problems. As shown in Fig. 2, by cascading the image unicoder, text unicoder, and cross-modal decoder, we can perform two tasks seamlessly by only switching the working modes of unicoders. A cross-entropy loss is applied on top of the cross-modal decoder to maximize the conditional likelihood of the ground-truth token under the forward autoregressive factorization.

$$\mathcal{L}_{\text{I2T}} = -\sum_{t=1}^T \log P_\theta(y_t|y_{<t}, I), \ \mathcal{L}_{\text{T2I}} = -\sum_{t=1}^T \log P_\theta(x_t|x_{<t}, T), \tag{2}$$

where $y$ and $x$ denote text and image tokens respectively.

**Classifier-Free Guidance for T2I.** Following Yu et al. (2022b); Chang et al. (2023); Ramesh et al. (2021), we employ classifier-free guidance (CFG) (Ho & Salimans, 2022) in text-to-image generation. To be more specific, in training, we randomly mask conditioning vectors, *i.e.*, input text tokens, by certain possibility (10% in our implementation). In inference, we compute two predictions: conditional one $I(z, T)$ and unconditional one $I(z)$, which only differ in text input: conditional prediction $I(z, T)$ has original text tokens as input while the input text of unconditional prediction $I(z)$ is fully masked. Then we linearly interpolate the $I(z, c)$ and $I(z)$ to obtain the final generated image:

$$I = I(z) + \alpha(I(z, T) - I(z)), \tag{3}$$

where $\alpha$ is a hyperparameter to adjust the scale of classifier-free guidance, and we set $\alpha$=2.0 in default.

**Final Loss.** In the end, we add those three losses up to optimize the model end-to-end.

$$\mathcal{L}_{\text{CoBIT}} = \lambda_{\text{Con}}\mathcal{L}_{\text{Con}} + \lambda_{\text{I2T}}\mathcal{L}_{\text{I2T}} + \lambda_{\text{T2I}}\mathcal{L}_{\text{T2I}} \tag{4}$$

where $\lambda_{\text{Con}}$, $\lambda_{\text{I2T}}$, $\lambda_{\text{T2I}}$ denote corresponding scalar coefficients for contrastive, I2T and T2I loss. In default, we set $\lambda_{\text{T2I}} : \lambda_{\text{I2T}} : \lambda_{\text{Con}} = 1 : 0.2 : 0.1$.

**Scaling.** As shown in Tab.1, we start from CoBIT-Base, and scale it up, w.r.t. both number of layers and model dimension, to obtain CoBIT-Large with around 1B parameters.

**Initialization.** In previous text-to-image generation models (Yu et al., 2022b; Rombach et al., 2022; Saharia et al., 2022), the text feature extractor is usually initialized by a pre-trained text model. Correspondingly, in CoBIT, we also initialize the text unicoder with another pre-trained text uni-modal decoder from CoCa (Yu et al., 2022a), while leaving the image unicoder and cross-modal decoder trained from scratch. In Sec. 4.4, we compare it with training all from scratch and find the initialization indeed helps.

## 3.4 COMPARISON WITH OTHER UNIFIED WORKS

For a clearer comparison with other unified works, we provide a detailed explanation of CoBIT *v.s.* recent unified diffusion-based and auto-regressive models, such as Versatile Diffusion (Xu et al., 2022), CoDi (Tang et al., 2023), Hu et al. (2022), UniDiffuser (Bao et al., 2023), Unified-IO (Lu et al., 2022), OFA (Wang et al., 2022b), BEIT-3 (Wang et al., 2022c) and CM3Leon (Yu et al., 2023) in Appendix 6.5.

# 4 EXPERIMENTS

In this section, we first describe the pre-training details (Sec. 4.1). Following, Sec. 4.2 and Sec. 4.3 introduce the primary results of zero-shot and fine-tuning evaluation, respectively. Both evaluations examine three capacities: (1) visual understanding, (2) image captioning and multimodal understanding, (3) text-to-image content creation. Lastly, Sec. 4.4 brings ablation.

## 4.1 PRE-TRAINING DETAILS

**Data.** CoBIT is designed to be pre-trained with image-text data. For contrastive loss and I2T loss, we use a mixture of ALIGN dataset (Jia et al., 2021), and JFT-4B dataset (Zhai et al., 2022a) where category names are transformed into texts by prompts as in Pham et al. (2021). Differently, for T2I generation, we found that the short text in JFT-4B is less informative for generating the image as extensive descriptions of visual details are important. Instead, we replace JFT with WebLI dataset (Chen et al., 2022), and mix it with ALIGN for T2I generation loss. We further perform de-duplication, as in Jia et al. (2021); Zhai et al. (2022b), to remove the examples close to downstream tasks. In the end, we obtain 1.1B pairs from ALIGN dataset, 162M pairs from WebLI dataset, and 4B pairs from JFT-4B dataset.

**Optimization.** CoBIT is implemented using Pax (Team, 2023), a Jax-based framework. Within each batch, for optimizing T2I loss, we sample 1,024 image-text pairs from a mixture of ALIGN and WebLI datasets, and for optimizing contrastive and I2T losses, we sample 30,720 image-text pairs from a mixture of ALIGN and JFT datasets. In total, the batch size is 31,744. We use the Adafactor (Shazeer & Stern, 2018) optimizer with $\beta_1 = 0.9$, $\beta_2 = 0.96$ and a weight decay of 0.045. As for the learning rate schedule, we warm it up to 4.5e-5 in the first 5,000 steps and then use an exponential decay starting from the step of 85,000. In total, models are pre-trained for 1M steps and CoBIT-Base/CoBIT-Large takes around 12 days on 256/512 CloudTPUv4 chips. Then, following Radford et al. (2021); Jia et al. (2021), we further pre-train our models for 50k steps with 576x576 high-resolution raw images as input in image encoding. The image input to ViT-VQGAN, *i.e.*, image for decoding, is kept at 256x256 resolution.

## 4.2 ZERO-SHOT EVALUATION ON DOWNSTREAM TASKS

We evaluate CoBIT on 5 representative tasks, and compare CoBIT against CLIP (Radford et al., 2021), ALIGN (Jia et al., 2021), FILIP (Yao et al., 2021), Florence (Yuan et al., 2021), (Yu et al., 2022a), ZeroCap (Tewel et al., 2021), SimVLM (Wang et al., 2021), VLKD (Dai et al., 2022), Parti (Yu et al., 2022b), LDM (Stable Diffusion)-1.4B (Rombach et al., 2022), Flamingo-3B (Alayrac et al., 2022), DALL-E 2 (Ramesh et al., 2022), Versatile Diffusion (Xu et al., 2022), CoDi (Tang et al., 2023) and CM3Leon (Yu et al., 2023).

**Zero-shot Image Classification.** We follow the standard evaluation protocols as in CLIP, ALIGN, *etc* (details in Appendix 6.4.2). As shown in Tab. 2, compared to models with similar scales, in ImageNet (Russakovsky et al., 2015), CoBIT-Large can achieve 82.7%, outperforming strong baselines such as CLIP and ALIGN. In practice, we find batch size difference also profoundly affects models' performance, which may contribute to the 2% discrepancy between the performance of CoBIT-Large and CoCa-Large as CoCa's batch size is 64k while ours is appendixonly.

**Zero-shot Image-Text Retrieval.** The image and text feature extraction process is the same as zero-shot image classification. Flick (Plummer et al., 2015) and MS-COCO (Lin et al., 2014) are used for evaluation. In Tab. 2, within comparable scales, CoBIT-Large can outperform the previous best model CoCa-Large in 5 out of 8 metrics and is ranked the second best in another two metrics.

**Zero-shot Image Captioning.** Since CoBIT is already pre-trained with image-to-text generation loss on noisy image-text data, we directly evaluate it on zero-shot image captioning. As in Tab. 2, in MS-COCO, CoBIT-Base/CoBIT-Large can achieve 43.0/44.8 CIDEr score, surpassing SimVLM by

Table 2: **Zeroshot Evaluation** of CoBIT against previous image-text models. The models in gray background have ≫ 1B parameters while others in white background have ≲ 1B parameters. For models with ≲ 1B parameters, we highlight the best score in **bold+underline** and the second-best score in underline. Understd. is the abbreviation of Understanding.

| Model | Image Understd. | Image-Text Understd. | | | | | | | | | Content Creation |
|---|---|---|---|---|---|---|---|---|---|---|---|
| | ImageNet Classification | Flickr Retrieval | | | | MS-COCO Retrieval | | | | MS-COCO Captioning | MS-COCO T2I Generation |
| | | Image→Text | | Text→Image | | Image→Text | | Text→Image | | | |
| | Acc(%) | R@1 | R@5 | R@1 | R@5 | R@1 | R@5 | R@1 | R@5 | CIDEr | FID (↓) |
| CLIP | 76.2 | 88.0 | 98.7 | 68.7 | 90.6 | 58.4 | 81.5 | 37.8 | 62.4 | - | - |
| ALIGN | 76.4 | 88.6 | 98.7 | 75.7 | 93.8 | 58.6 | 83.0 | 45.6 | 69.8 | - | - |
| FILIP | 78.3 | 89.8 | 99.2 | 75.0 | 93.4 | 61.3 | 84.3 | 45.9 | 70.6 | - | - |
| Florence | 83.7 | 90.9 | 99.1 | 76.7 | 93.6 | 64.7 | **85.9** | 47.2 | 71.4 | - | - |
| CoCa-Large | **84.8** | 91.4 | **99.2** | 79.0 | 95.1 | **65.4** | 85.6 | 50.1 | 73.8 | - | - |
| ZeroCap | - | - | - | - | - | - | - | - | - | 14.6 | - |
| SimVLM | - | - | - | - | - | - | - | - | - | 32.2 | - |
| VLKD | - | - | - | - | - | - | - | - | - | 58.3† | - |
| Parti-350M | - | - | - | - | - | - | - | - | - | - | 14.10 |
| Parti-750M | - | - | - | - | - | - | - | - | - | - | 10.71 |
| LDM (SD)-1.4B | - | - | - | - | - | - | - | - | - | - | 12.63 |
| Coca-2B | 86.3 | 92.5 | 99.5 | 80.4 | 95.7 | 66.3 | 86.2 | 51.2 | 74.2 | - | - |
| Make-A-Scene | - | - | - | - | - | - | - | - | - | - | 11.84 |
| Versatile Diffusion | - | - | - | - | - | - | - | - | - | - | 11.10 |
| CoDi | - | - | - | - | - | - | - | - | - | - | 11.26 |
| DALL-E 2 | - | - | - | - | - | - | - | - | - | - | 10.39 |
| CM3Leon-7B | - | - | - | - | - | - | - | - | - | - | 10.82 |
| Parti-20B | - | - | - | - | - | - | - | - | - | - | 7.23 |
| CoBIT-Base | 79.4 | 89.5 | 98.4 | 76.5 | 94.3 | 62.1 | 83.5 | 47.3 | 72.3 | 43.0 | 10.35 |
| CoBIT-Large | 82.7 | **91.5** | 99.1 | **79.9** | **95.3** | 65.1 | 85.5 | **50.3** | 74.2 | **44.8** | **9.37** |

10.8/12.6. It's noted that the models with †, *e.g.*, Flamingo, VLKD, have much higher scores than others because they reuse a pre-trained large language model as a decoder that inherits strong text generation ability.

**Zero-shot Text-to-Image Generation.** We follow the standard evaluation process as in Parti and DALL-E (detail in Appendix 6.4.3). As we can see in Tab.2, CoBITs can beat specialized models with comparable scales, and CoBIT-Large can achieve an impressive FID of 9.37 which outperforms some models with larger scales by a significant margin, *e.g.*, DALL-E 2/Make-A-Scene with 3.5B/4B parameters.

## 4.3   Fine-tuning on Downstream Tasks

To demonstrate the transferability of CoBIT, we further conduct linear probing or fine-tuning on multiple downstream tasks. Besides the existing methods we mentioned in the previous section, we also compare UNITER (Chen et al., 2019), VinVL (Zhang et al., 2021b), CLIP-ViL (Shen et al., 2021), OFA (Wang et al., 2022b), X-LXMERT (Huang et al., 2021) and PALI (Chen et al., 2022). The detailed hyperparameters of these tasks are shown in Tab. 7.

**Linear Probing on ImageNet.** Following CLIP, we linear probe CoBIT by fixing all parameters of the image unicoder and only training a linear classifier on top for image recognition. CoBIT-Large can outperform CLIP and ALIGN by around 1%.

**Image-Text Understanding.** We categorize VQA (Antol et al., 2015), SNLI-VE (Xie et al., 2019) and image captioning into tasks requires image-text understanding. We fine-tune all parameters of CoBIT and evaluate it on the val/test set.

*Captioning.* In fine-tuning, CoBIT computes caption predictions in the same way as zero-shot image captioning in Sec. 4.2. In Tab. 3, we can see the CoBIT can achieve a competitive CIDEr score against other models. It's noted that some works (Wang et al., 2022b) additionally apply task-specific tricks such as CIDEr optimization, but for a fair comparison, we only present their results with plain cross-entropy loss.

*VQA.* We follow prior works (Wang et al., 2021; Yu et al., 2022a) to setup VQA fine-tuning (Detail in Appendix 6.4.4). As shown in Tab. 3, CoBIT can achieve satisfactory performance compared with other VLP models.

Table 3: **Fine-tuning Evaluation** of CoBIT against previous image-text models. PT. denotes pretrained, and Scratch denotes trained from scratch. [†]OFA incorporates images and text in its input, while others only use image one.

| Model | Visual Backbone | Image Understd. ImageNet Linear Probing | Image-Text Understd. VQA test-dev | test-std | SNLI-VE dev | test | MS-COCO Captioning (CIDEr) | Content Creation MS-COCO T2I Generation (FID↓) |
|---|---|---|---|---|---|---|---|---|
| CLIP | Scratch | 85.4 | - | - | - | - | - | - |
| ALIGN | Scratch | 85.5 | - | - | - | - | - | - |
| UNITER | Faster-RCNN | - | 73.8 | 74.0 | 79.4 | 79.4 | - | - |
| VinVL | Faster-RCNN | - | 76.5 | 76.6 | - | - | 130.8 | - |
| CLIP-ViL | CLIP | - | 76.5 | 76.7 | 80.6 | 80.2 | 134.2 | - |
| ALBEF | PT. ViT | - | 75.8 | 76.0 | 80.8 | 80.9 | - | - |
| BLIP | PT. ViT | - | 78.3 | 78.3 | - | - | 136.7 | - |
| SimVLM | PT. ResNet | - | 80.0 | 80.3 | **86.2** | **86.3** | 143.3 | - |
| OFA | PT. ResNet | - | **82.0** | **82.0** | 91.0 [†] | 91.2[†] | **145.3** | 10.5 |
| X-LXMERT | Faster-RCNN | - | - | - | - | - | 122.6 | 29.9 |
| Unified-IO$_{XL}$-2.9B | Scratch | - | - | 77.9 | 91.1 | - | 122.3 | - |
| CoDi | CLIP | - | - | - | - | - | 149.9 | 3.22 |
| CoCa-2.1B | Scratch | - | 80.0 | 80.3 | 87.0 | 87.1 | 143.3 | - |
| BEIT3-1.9B | Scratch | - | 84.2 | 84.0 | - | - | 147.6 | - |
| PALI-17B | PT. ViT | - | 84.3 | 84.3 | - | - | 149.1 | - |
| Parti-20B | - | - | - | - | - | - | - | 3.22 |
| CoBIT-Base | Scratch | 83.48 | 76.3 | 76.6 | 85.4 | 85.4 | 135.4 | 5.06 |
| CoBIT-Large | Scratch | **86.44** | 77.9 | 78.3 | **86.2** | 86.0 | 139.5 | **4.62** |

***SNLI-VE.*** Similar to fine-tuning VQA, we extract the final token output feature of the cross-modal decoder and apply a linear classifier on top to predict the three relations. As shown in Tab. 3, CoBIT can outperform strong VLP models and achieve superior performance. Note that other models, including CoBIT only use image premises as inputs, but OFA incorporates both image and text premises in its input.

**Text-to-Image Generation.** Following Parti and DALL-E, we fine-tune CoBIT on MS-COCO training set and evaluate the FID score on the sampled appendixtest set. Compared with zero-shot performance, fine-tuning on CoBIT-Base/CoBIT-Large further reduces the FID from 10.35/9.37 to 5.06/4.62, outperforming models of comparable scales.

## 4.4 ABLATION

This section comprehensively ablates the design choices in CoBIT. Most ablation experiments are conducted on CoBIT-Base with a reduced batch size and a shrunken training schedule (See detail in Appendix 6.4.5). We select the following representative tasks: zero-shot ImageNet Classification (ZS IN.) for image understanding, VQA (fine-tuned) or zero-shot Captioning (ZS Cap.) for multimodal understanding, and zero-shot text-to-image generation (ZS IG.) for image generation.

**Training Objectives.** We ablate the existence of three training objectives: contrastive loss, I2T loss, and T2I loss, and study how they affect each other. The result is shown in Tab. 4. We can obtain several interesting observations: (1). By comparing the first and last rows, it is found *cross-modal generation objectives can improve image understanding a bit on top of contrastive loss*. The zero-shot ImageNet accuracy is improved by 0.3%. (2) Comparing the second, third, and fourth rows, we see *two generations losses, i.e., I2T loss and T2I loss, contradict each other a little bit. But it's also promising because joint training essentially saves half of the parameters compared with using an ensemble of two separate models*. (3) From the fourth row and fifth row, we can see *contrastive loss improves vision-language understanding while it doesn't influence image generation*. Overall, we demonstrate the feasibility of harmoniously unifying three fundamental objectives in one framework.

**Loss Weight.** Given three objectives, we ablate different weights for them and select the best one as the default configuration for all experiments. Please see Appendix 6.2.

**Unicoder vs. Encoder.** In previous Vision-Language works (Wang et al., 2021; Yu et al., 2022a; Chen et al., 2022), encoder-decoder has been a de facto pipeline, where encoder encodes image/text features and cross-modal decoder fuses them and perform generation. Differently, we propose unicoder to replace encoder, which can both encode and decode unimodal representations with shared parameters. Here, we ablate image and text unicoders against image and text encoders. In the Ap-

Table 4: Ablation on three objectives. Con. is contrastive loss.

| Objectives | | | Evaluation | | |
|---|---|---|---|---|---|
| Con. | T2I | I2T | ZS IN. | VQA | ZS IG. (↓) |
| ✓ | - | - | 70.8 | - | - |
| - | ✓ | - | - | - | 12.6 |
| - | - | ✓ | - | 68 | - |
| - | ✓ | ✓ | - | 65.4 | 13.2 |
| ✓ | ✓ | ✓ | 71.1 | 66.9 | 13.3 |

Table 5: Ablation on unicoder vs. encoder.

| Module | | Evaluation | | |
|---|---|---|---|---|
| Image | Text | VQA | ZS Cap. | ZS IG. (↓) |
| Encoder | Encoder | 65.9 | 32.9 | 13.8 |
| Unicoder | Encoder | 66.5 | 36.9 | 13.38 |
| Encoder | Unicoder | 67.8 | 35.0 | 13.67 |
| Unicoder | Unicoder | 66.9 | 37.9 | 13.31 |

Table 6: Ablation on initialization.

| | Init. Text from CoCa | From Scratch |
|---|---|---|
| ZS IN. | 75.35 | 75.02 |
| VQA | 68.48 | 68.55 |
| ZS IG. (↓) | 11.42 | 11.63 |

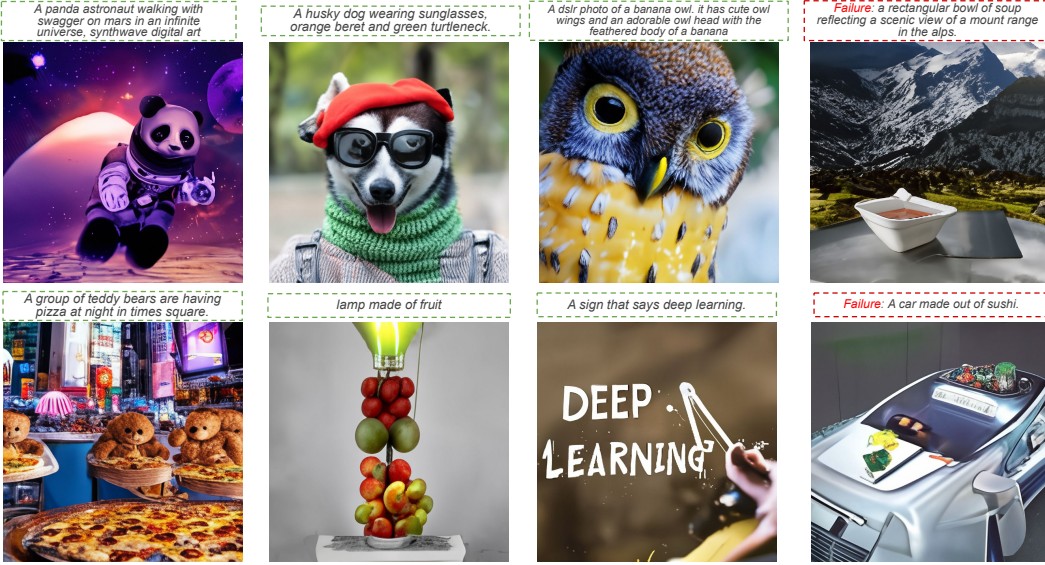

Figure 3: Qualitative results of zero-shot text-to-image generation from CoBIT-Large with both good and failed cases.

pendix 6.6, we put a diagram Fig. 4 to illustrate how the compared encoder-only models work, and explain the number of parameters of those two designs. As shown in Tab. 5, either image unicoder or text unicoder can improve over encoders, and applying them together brings the best trade-off for both image generation and multimodal understanding.

**Pre-training Data.** The ablation of three different pre-training datasets is detailed in Appendix 6.3.

**Train From Scratch.** As mentioned in Sec. 3.3, we initialize the text unicoder with a pre-trained unimodal text decoder from CoCa. Here, we also attempt to train all from scratch. In this comparison, all models are trained with non-shrunken batch size to mitigate the possible gap due to the much larger batch size of CoCa. In Tab. 6, loading pre-trained weight from CoCa improves zero-shot Imagenet recognition and text-to-image generation by 0.3% and 0.2, which is a small margin. Also, it doesn't even improve VQA. This comparison verifies the do-ability of training CoBIT all from scratch without hurting much performance.

## 4.5 VISUALIZATION

We visualized good and failed generated images of CoBIT-Large using the prompts from PartiPrompt (Yu et al., 2022b). As in Fig. 3, CoBIT can generate high-quality, broadly capable, open-domain images based on text. As for failed cases, we can see CoBIT misunderstands "A car made out of Sushi" as "A car with Sushi on top", also CoBIT fails to generate the reflection of mountains in the bowl of soup. More visualization and analysis are in Fig. 5.

## 5 CONCLUSION

We present a VL foundation model, CoBIT, which unifies three objectives: cross-modal contrastive learning, image-to-text generation, and text-to-image generation. The model is trained on large-scale noisy web-crawled image-text and image annotation data. CoBIT achieves strong zero-shot and transferable capacities of unimodal visual understanding, image-text matching, image-text understanding, and text-to-image content creation.

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

| | ImageNet Linear Probe | VQA | SNLI-VE | MS-COCO Captioning | MS-COCO T2I Generation |
|---|---|---|---|---|---|
| Optimizer | SGD | Adafacter | | | |
| Gradient Clip | 1.0 | | | | |
| LR decay schedule | Cosine Schedule to zero | Linear Schedule to zero | | | Exponential Schedule to zero |
| RandAugment | 2, 5 | 1, 10 | | | None |
| Training Step | 225k | 100k | 50k | 15k | 100k |
| Warm-Up Step | 0 | 1000 | 1000 | 500 | 1000 |
| Batch Size | 512 | 32 | 128 | 128 | 256 |
| Learning Rate | 3.2 | 1e-5 | 5e-5 | 5e-6 | 1e-5 |
| Weight Decay | 0.0 | 0.1 | 0.1 | 0.01 | 0.045 |

Table 7: Hyper-parameters used in the multimodal experiments.

# 6 APPENDIX

## ACKNOWLEDGEMENT

We would like to thank Prof. Shih-Fu Chang, Luowei Zhou, Long Chen for constructive discussion, Zirui Wang for help with downstream fine-tuning, Huiwen Chang for proofreading, and Laurent El Shafey for infra support.

## 6.1 LIMITATIONS & BROADER IMPACT

**Limitations.** Although CoBIT unifies contrastive loss, text-to-image generation (T2I) loss, and image-to-text (I2T) generation loss, from ablation experiments, we can find that T2I and I2T objectives contradict each other a little bit. We hypothesize that it's because two generations require some fine-grained knowledge that is specific to each modality. We will leave it as a future direction.

**Broader Impact.** Models such as Stable Diffusion, DALL-E, Parti, CoBIT are trained on large and noisy image-text datasets that inevitable include potential biases toward people of different backgrounds. The bias can arise because the dataset's image and text samples might not be representative of the real-world population and could inadvertently promote certain stereotypes. Also, because the generated images of CoBIT are in high quality, the concern of abusing the model to create and spread manipulated data (deepfake, political manipulation, etc.) does exist.

## 6.2 ABLATION ON LOSS WEIGHT

Table 8: Ablation on weights of three losses. Con. denotes contrastive loss. T2I denotes text-to-image generation loss. I2T denotes image-to-text generation loss. ZS IN. denotes zero-shot ImageNet classification. ZS IG. denotes zero-shot text-to-image generation on MS-COCO, which is evaluated by FID and lower FID is better.

| Weights | | | Evaluation | | |
|---|---|---|---|---|---|
| Con. | T2I | I2T | ZS IN. | VQA | ZS IG. ($\downarrow$) |
| - | 0.1 | 1 | - | 63.2 | 13.17 |
| - | 0.2 | 1 | - | 65.4 | 13.24 |
| - | 1 | 1 | - | 67.8 | 16.33 |
| 0.1 | 0.2 | 1 | 71.1 | 66.9 | 13.31 |
| 0.4 | 0.2 | 1 | 71.2 | 66.5 | 13.92 |

Given three objectives, we ablate different weights for them and select the best one as the default configuration for all experiments. We start with T2I and I2T first: given the weight of T2I fixed to

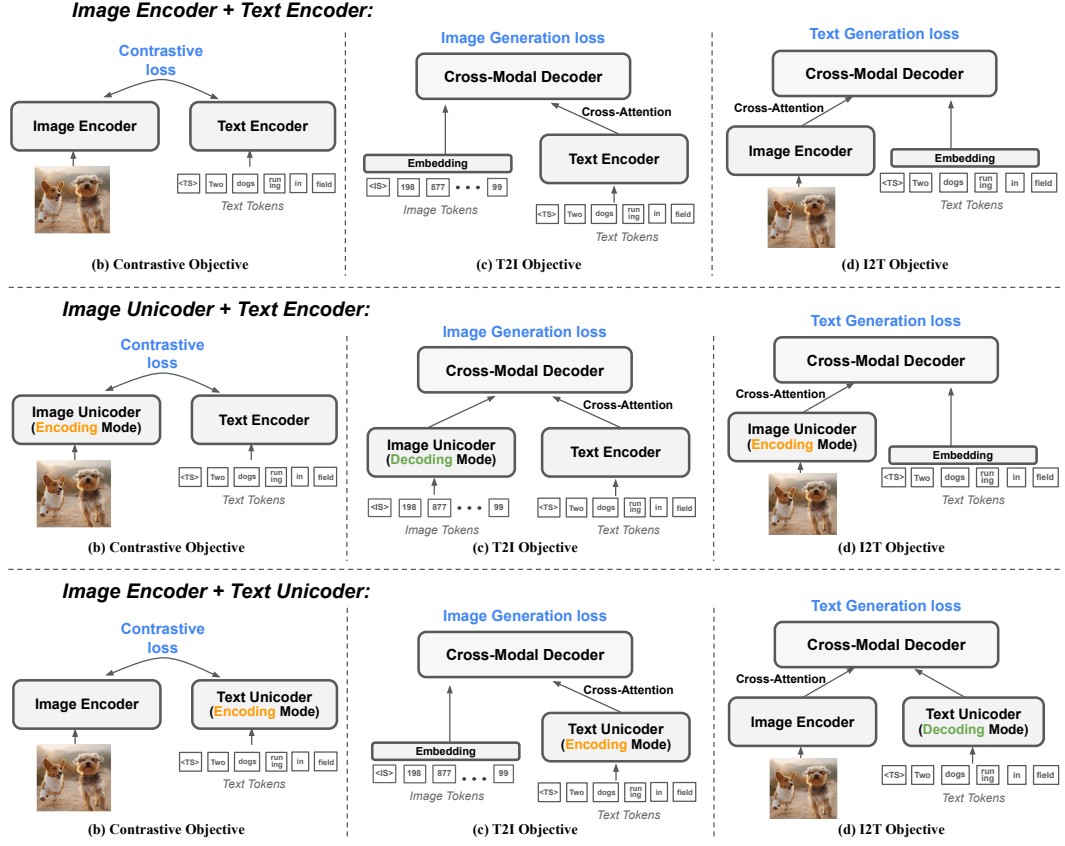

Figure 4: Diagram of three compared models in the ablation of **unicoder vs. encoder**. **Top**: Replacing both image unicoder and text unicoder with image encoder and text encoder respectively. **Middle**: Replacing text unicoder with text encoder while keeping image unicoder. **Bottom**: Replacing image unicoder with image encoder while keeping text unicoder.

1, we ablate the loss of I2T. Then given T2I and I2T loss both fixed, the weight of contrastive loss is ablated. As we can see in Tab. 9, a high weight of I2T such as 1 will hurt the image generation heavily but also improve VQA. On the other hand, a high weight of contrastive loss like 0.4 will not essentially improve image recognition and hurts both VQA and image generation. Overall, we chose Con.:T2I:I2T = 0.1:0.2:1 as our default setting, as it achieves a good trade-off between three losses.

## 6.3 ABLATION ON PRE-TRAINING DATA

Table 9: Ablation on weights of three pre-training datasets. ZS IN. denotes zero-shot ImageNet classification. LP. means linear probing. ZS IG. denotes zero-shot text-to-image generation on MS-COCO, which is evaluated by FID and lower FID is better.

| Datasets | Evaluation | | | |
|---|---|---|---|---|
| | ZS IN. | LP. IN. | VQA | ZS IG. ($\downarrow$) |
| JFT | 71.6 | 81.4 | 64.8 | 14.6 |
| ALIGN | 70.9 | 81 | 67.2 | 13.8 |
| WebLI | 70.0 | 80.2 | 66.2 | 13.4 |

Here we ablate the three pre-training datasets. To make a fair comparison, the batch size is kept the same, the training is conducted for 200k steps on base model. As we can see in Tab. 9, JFT is beneficial to classification tasks, focusing on basic and precise semantics of image; WebLI has higher-quality image data and is specifically beneficial to text-to-image generation; ALIGN is relatively noisy but covers broad semantics. That's the reason why we mixed them for a more balanced training.

## 6.4 DETAILS OF EXPERIMENTS

### 6.4.1 HYPERPARAMETERS IN FINE-TUNING

In Tab. 7, we present the hyperparameters we used in fine-tuning/linear probing of CoBIT.

### 6.4.2 ZERO-SHOT IMAGE CLASSIFICATION

we apply the same set of prompts to transfer labels into sentences, such as "a photo of {class}". Similar to the contrastive loss computed in Sec. 3.3, we input raw image/text into image/text unicoders in encoding mode to obtain the global image and text features. Then, we compute their similarity to match images and labels.

### 6.4.3 ZERO-SHOT TEXT-TO-IMAGE GENERATION

In decoding, we employ Top-K sampling to sample 16 images for each text and use a reranker to select the best image for evaluation. Following the de facto process, we compute FID score (Heusel et al., 2017) on MS-COCO appendixdata (lower FID is better).

### 6.4.4 VQA FINE-TUNING

we use the VQA v2, and the task is formulated as a classification problem over 3,129 most frequent answers in the training set. To accomplish this, the raw image is fed into the image unicoder using encoding mode, while the question is processed by the text unicoder in decoding mode. Subsequently, the cross-modal decoder utilizes the text decoding features as input and cross-attends to the encoded image features. The final token output feature of the cross-modal decoder is considered the fused global feature. To predict the answer, a linear classifier is trained on top of this feature.

### 6.4.5 SETUP OF ABLATION TRAINING

Specifically, the total batch size is 4,352, containing 4,096 for contrastive and I2T loss and 256 for T2I loss, and the total training step is 200k without high-resolution pre-training.

## 6.5 DETAILED COMPARISON WITH OTHER UNIFIED WORKS.

*v.s.* **Unified Diffusion-based Models.** Some recent works utilize diffusion models to jointly learn text-to-image and image-to-text learning, such as Versatile Diffusion (Xu et al., 2022), CoDi (Tang et al., 2023), Hu et al. (2022), UniDiffuser (Bao et al., 2023). Although they work well in image generation, they tend to perform worse in text generation and fail to handle image-text understanding tasks such as VQA, retrieval, etc. Moreover, they all initialize from Stable Diffusion, while CoBIT is mostly trained from scratch and learns superior image understanding capability.

*v.s.* **Unified Auto-Regressive Models.** Unified-IO (Lu et al., 2022), OFA (Wang et al., 2022b) and CM3Leon (Yu et al., 2023) also train text-to-image and image-to-text jointly. However, they use a plain decoder or encoder-decoder model without considering the contrastive alignment as in CoBIT. BEIT-3 (Wang et al., 2022c) conducts the image-to-text generation by mask-and-reconstruction with an encoder model but could not handle the text-to-image generation task.

## 6.6 ILLUSTRATION OF REPLACING UNICODERS WITH ENCODERS IN COBIT

In Sec.4.4, we ablate **Unicoder vs. Encoder** and demonstrate the effectiveness of proposed unicoders. In Fig. 4 we show the diagram of using image and text encoders, image encoder+text unicoder, and image unicoder+text encoder. As we can see, encoders can only encode visual or textual

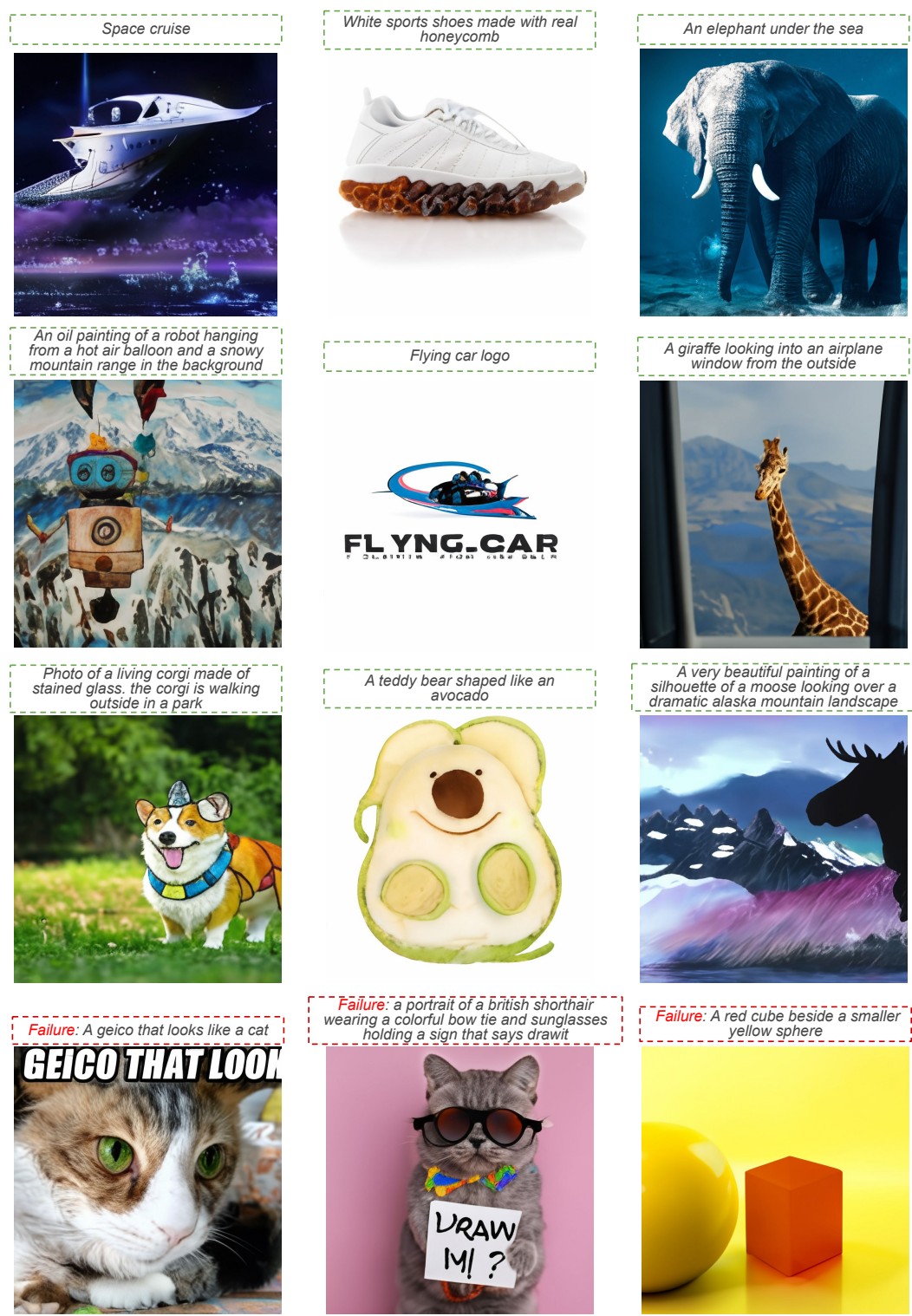

Figure 5: Qualitative results of zero-shot text-to-image generation from CoBIT-Large with both good and failed cases.

features while unicoders can perform both encoding and decoding, which shares the knowledge and boosts the generation result as shown in previous ablation. It's noted that in pre-training, unicoder doesn't add extra parameters compared to encoders because encoding and decoding in unicoder reuse the same set of parameters; In the finetuning of text-to-image and image-to-text tasks, the uni-

coder design indeed brings more parameters than encoder. Therefore, we mainly evaluate zero-shot captioning and zero-shot image generation in this ablation (Tab. 5) to eliminate the difference of parameter numbers.

## 6.7 MORE VISUALIZATION

In Fig. 5, We attach more visualization of CoBIT-Large on zero-shot text-to-image generation with novel prompts in PartiPrompts Yu et al. (2022b). For better visualization when zoom-in, we employ Sahak et al. (2023) as the super-resolution module to upsample generated 256x256 images to 1024x1024 images. It's noted that when computing FID, we still use 256x256 images and the high-resolution ones are only used for visualization. In failed cases, we find that: (1) CoBIT sometimes messes up the size attributes of two objects. For example, in the last example, yellow sphere ought to be smaller. (2) CoBIT sometimes couldn't render the details of words in text very well. In the second last example, "DRAWIT" is rendered as "DRAWMI?". (3) CoBIT occasionally misunderstands the text. In the third last example, we expect a geico that looks like a cat whereas CoBIT first renders "GEICO THAT LOOK" then generates a cat. It's indeed a new way to interpret the text but not the desired way of humans.

