# OpenReview forum: "CoBIT: A Contrastive Bi-directional Image-Text Generation Model"
_ICLR.cc/2024/Conference — ICLR 2024 poster_

### Official Review · Reviewer_vaSs · 2023-10-30

**Soundness:** 3 good
**Presentation:** 3 good
**Contribution:** 3 good
**Rating:** 6
**Confidence:** 4

**Summary:**

The paper presents a novel model named Contrastive Bi-directional Image-Text generation model (CoBIT) aimed at unifying three pre-training objectives, namely, cross-modal contrastive learning, image-to-text generation, and text-to-image generation, under a singular framework. This unification is achieved through a unique unicoder-decoder architecture that houses an image unicoder, a text unicoder, and a cross-modal decoder. The unicoders are capable of toggling between encoding and decoding roles based on the task at hand, thereby promoting shared knowledge which is advantageous for both image-to-text and text-to-image generations. The authors claim that this model architecture allows for superior performance in various tasks like image understanding, image-text understanding, and text-based content creation, with a significant highlight on its effectiveness in zero-shot scenarios. The model's efficiency is validated through a series of comprehensive experiments, demonstrating an impressive performance against existing models in the field of Vision-and-Language (VL).

**Strengths:**

- Unified Framework: CoBIT effectively brings together three prevalent pre-training objectives under a single framework, which could potentially lead to a more holistic understanding and representation of image-text pairs.

- Flexible Architecture: The unicoder-decoder structure is innovative and allows for flexibility and shared knowledge, which is beneficial for multiple generation tasks.

- Zero-shot Performance: The model demonstrates high accuracy and superior performance in zero-shot scenarios across various tasks like image understanding, image-text retrieval, image captioning, and text-to-image generation.

- Parameter Efficiency: The paper highlights excellent parameter efficiency, as the same set of Transformer parameters are utilized for both encoding and decoding tasks, which is a crucial factor considering the computational resources.

**Weaknesses:**

- Assumed Synergy: The core premise of CoBIT hinges on the idea of a symbiotic relationship between image-to-text and text-to-image generation tasks. However, the paper doesn't thoroughly investigate or justify the assumed synergy. It's crucial to establish a theoretical foundation for this assumption, or the unified framework might not hold in different contexts or datasets.

- Objective Conflicts: While unifying different objectives under a single framework is innovative, it poses a risk of conflicting objectives that might detract from optimizing each task individually. The paper acknowledges this to some extent but doesn't provide a robust solution to mitigate potential conflicts.

- Evaluation Scope: The evaluation primarily focuses on showcasing the model's strengths, with a less thorough investigation into the model’s weaknesses or failure modes. A more balanced evaluation, including a deeper exploration of where the model falls short, would provide a more comprehensive understanding of the model's capabilities and limitations.

**Questions:**

- Theoretical Justification: A deeper theoretical analysis of the assumed synergy between image-to-text and text-to-image generation tasks could strengthen the premise of the unified framework.

- Conflict Mitigation Strategies: Developing and incorporating strategies to mitigate the potential conflicts between different objectives could help in achieving a more balanced optimization across all tasks.

- Failure Analysis: Conducting a thorough failure analysis to identify the model's weaknesses and understanding its behavior under different conditions or limitations could provide a clearer path for future improvements.

---

> ### Author Response · Authors · 2023-11-21
> **Response by Author(s)**
>
> > **W1: Theoretical Justification of Assumed Synergy.**
>
> **Answer**: We really appreciate your excellent suggestion. Based on the ablation in Sec. 4.4, we would like to provide further analysis of the mutual effects of three different losses:
>
>
> * **Finding**: Cross-modal generation objectives can improve image understanding a bit on top of contrastive loss. **Analysis**: Either T2I objective or I2T objective focuses on cross-modal fine-grained semantics, which can explicitly facilitate the global alignment between two modalities.
>
> * **Finding**: Two generation losses, i.e., I2T loss and T2I loss, contradict each other a little bit. **Analysis**: The good signal is that those two don’t contradict that much and still can keep a comparable performance when jointly training. It’s promising because joint training essentially saves half of the parameters compared with using an ensemble of two separate models. On the other hand, those two losses don’t show mutual benefit in performance as we expected. We think the reason is that the granularity of image generation and text generation are different. A single word like cat in the text may correspond to a region in the image that contains hundreds of pixels and tens of image tokens.
>
> * **Finding**: Contrastive loss improves vision-language understanding while it doesn’t influence image generation. **Analysis**: A good global alignment enabled by contrast provides the base for better image-to-text generation. However, image generation requires precise local details beyond just correct semantics, which makes it hard to benefit from global alignment.
>
> As you can see, the mutual effects are kind of mixed, but the parameter efficiency of joint training always holds, i.e., the number of parameters in the ensemble of three specialized models is 2-3 times that of a jointly trained model.
>
> As for a more theoretical justification, we think existing works [1] about gradient harmony in multi-lingual model training might provide a new angle to our scenario. We two share the similarity in terms of multiple objectives, large data size, large model size, and intuitively mutual benefits of different objectives (vision is believed to be another language). Studying how the gradients of those three objectives interact should be more fundamental and theoretical, but also sophisticated. We would like to leave it as the next step.
>
> [1] Wang, Zirui, et al. "Gradient vaccine: Investigating and improving multi-task optimization in massively multilingual models." arXiv preprint arXiv:2010.05874 (2020).
>
> > **W2: Objective Conflicts Mitigation Strategies.**
>
> **Answer**: Thanks for your valuable suggestion. We do consider the design of the proposed Unicoder-Decoder an initial exploration of the conflict mitigation strategies. From the ablation result, when jointly training with different objectives, it can outperform conventional encoder-decoder models. Intuitively, it shares the knowledge between encoding a modality for cross-modality understanding and generating a modality from the other modality, which explicitly bridges both understanding loss and generation loss by sharing the parameter space. We think there is still plenty of space to explore and believe our findings can shed light on future works.
>
> > **W3: Weakness and failure analysis.**
>
> **Answer**: Thank you for pointing out. In Sec. 4.5 and Sec. 6.6, we conducted some failure analysis about image generation. Several observations were presented and we quote them here. We also included the visualization examples for each type of failure in the manuscript.
>
>
> * CoBIT sometimes messes up the size attributes of two objects.
>
> * CoBIT sometimes couldn’t render the details of words in text very well.
>
> * CoBIT occasionally misunderstands the text.
>
>
> Here we further add a brief failure analysis of image-to-text generation in zeroshot scenario:
>
> * CoBIT tends to generate short text, for example, one sentence or multiple phrases. This might be due to the training data distribution.
>
> * CoBIT sometimes cannot follow instructions but instead tends to complete the input text. Additionally, instruction-tuning should help.
>
> Based on the above points, we will provide a more detailed anaylsis to help future improvement in the camera-ready version.

---

> > ### Author Response · Authors · 2023-11-23
> > **The reviewer-author discussion will end within 4 HOURS**
> >
> > Dear Reviewer vaSs,
> >
> > The reviewer-author discussion period is nearing its conclusion in under 4 hours. If you have any further thoughts or comments, don't hesitate to share them with us. We sincerely thank you for your dedication and time in reviewing our submission.
> >
> > Warmest regards,
> > Author(s)

---

### Official Review · Reviewer_GAkS · 2023-10-30

**Soundness:** 3 good
**Presentation:** 3 good
**Contribution:** 2 fair
**Rating:** 5
**Confidence:** 4

**Summary:**

This paper proposes to unify three commonly used objectives in VL pre-training, namely cross-modal contrastive learning, image-to-text generation, and text-to-image generation, into one pretraining framework. In particular, the authors share the same parameters for Transformer in encoding and decoding. After pretraining on ALIGN, JFT-4B and WebLI datasets, the model shows superiority on image understanding, image-text understanding (retrieval, VQA, captioning) and content creation (text-to-image) tasks among models smaller than 1B parameters.

**Strengths:**

1.	This paper is well-written and easy to follow.
2.	Experimental results show good performance in several benchmarks.

**Weaknesses:**

1.	The whole pipeline is not novel and is more like an assembling of existing method/components. As mentioned by the authors, the unicoder is already used in other works  (Sec. 3.2). Three objectives for cross-modal learning are commonly used. Even though it is first joint learned, I cannot see the advantage of combining them. Although the authors claim that they should benefit from each other, it is intuitive that they have different focuses. For example, contrastive loss will benefit retrieval tasks. It could be better if the authors could have more insights on each objective’s advantages from their results.
2.	The advantages of the model over other models are not clear. There is no performance that shows fair comparison with other models. It could be the dataset used for pre-training, batch-size, pretraining epochs, etc.
3.	From the ablation studies in Table 4, 5 and 6, the gain of proposed joint training of three objectives and shared Transformer (namely unicoder) is rather small and not steady over all tasks. This could not convince me of the advantage of proposed method.
Overall, I think the insight and technical contribution of this paper are limited.

**Questions:**

•	Can you list the exact parameter sizes of each model in Table1,2,3? That should be more fair comparison.
•	Why are the tasks shown in ablation studies (Table 4,5,6) not consistent? What is the reason for lacking some tasks in each table?
•	Please check the weakness for more questions.

---

> ### Author Response · Authors · 2023-11-21
> **Response by Author(s) - 1/2**
>
> > **W1: Insights on each objective’s advantages.**
>
> **Answer**: Thank you for your valuable suggestion. Here we provide a more detailed explanation of each objective’s effects from their results.
>
> First of all, the parameter efficiency of joint training always holds, i.e., the number of parameters in the ensemble of three specialized models is 2-3 times that of a jointly trained model. Second, we don’t intend to make a point that those three objectives must benefit each other, but rather we want to serve as the first exploration about the relationship of those three objectives. In the following, we analyze the three findings we stated in Sec. 4.4 of original submission.
>
> * **Finding1**: Cross-modal generation objectives can improve image understanding a bit on top of contrastive loss. **Analysis**: Either T2I objective or I2T objective focuses on cross-modal fine-grained semantics, which can implicitly facilitate the global alignment between two modalities.
>
> * **Finding2**: Two generation losses, i.e., I2T loss and T2I loss, contradict each other a little bit. **Analysis**: The good sign is that those two don’t contradict that much and still can keep a comparable performance when jointly training. It’s promising because joint training essentially saves half of the parameters compared with using an ensemble of two separate models. On the other hand, those two losses don’t show mutual benefit in performance as we expected. We think the reason is that the granularity of image generation and text generation are different. A single word like cat in the text may correspond to a region in the image that contains hundreds of pixels and tens of image tokens.
>
> * **Finding3**: Contrastive loss improves vision-language understanding while it doesn’t influence image generation. **Analysis**: A good global alignment enabled by contrast provides the base for better image-to-text generation. However, image generation requires precise local details beyond just correct semantics, which makes it hard to benefit from global alignment.
>
> As you can see, the mutual effects are kind of mixed in terms of performance, but overall those three losses are compatible in the unified framework. We believe that there is plenty of space for better unifying these objectives. And our proposed Unicoder-Decoder provides an option for model design improvement.
>
> > **W2: Comparison w/ other models.**
>
> **Answer**: Thank you for your feedback. Comparing with other methods in an absolutely fair environment is not realistic because many works use their own private datasets, such as CLIP, Florence, DALLE, etc, and different models have different model sizes, batch sizes, and training steps.
> Since jointly learning cross-modal contrastive optimization, text-to-image generation, and image-to-text generation is a new setting, ours serves as an initial exploration and mainly studies how those three objectives impact each other, and which model structure is more effective and efficient (Unicoder-Decoder vs. Encoder-Decoder). On those two core problems, all our experiments and ablations are **conducted fairly with the same amount of data**. So the findings of our paper should be valid and useful to the community.
>
> > **W3: Inconsistent experimental results.**
>
> **Answer**: We apologize for the possible confusion and appreciate this opportunity to clarify the goal and contribution of our work.
>
> * **Joint training of three objectives** First of all, joint training is more efficient than separate training in terms of the number of parameters, as each separate trained model has its own parameters and in total three separate models’ parameter is 2-3x that of a jointly trained model. Second, we don’t intend to make a point that those three objectives must benefit each other, but rather we want to serve as the first exploration about the relationship of those three objectives. And we summarized three observations, as stated in Sec. 4.4:
>
> 	* Cross-modal generation objectives can improve image understanding a bit on top of contrastive loss
>
> 	* Two generations losses, i.e., I2T loss and T2I loss, contradict each other a little bit
>
> 	* Contrastive loss improves vision-language understanding while it doesn’t influence image generation.
>
> Overall, we demonstrate the feasibility of compatibly unifying three fundamental objectives in one framework without significant contradiction. We understand that there is still space for better unifying these objectives, and our proposed Unicoder-Decoder is an improvement from the model design.
>
> * **Unicoder-Decoder** We arguably believe the improvement of Unicoder-Decoder over conventional Encoder-Decoder is stable and significant on average. On all three tasks, it’s all better and the average improvement is 6.7% relatively (first row vs. last row of Tab. 5).

---

> ### Author Response · Authors · 2023-11-21
> **Response by Author(s) - 2/2**
>
> > **Questions: Can you list the exact parameter sizes of each model in Table1,2,3? That should be more fair comparison. • Why are the tasks shown in ablation studies (Table 4,5,6) not consistent? What is the reason for lacking some tasks in each table? • Please check the weakness for more questions.**
>
> **Answer**:  In Tab1, we already list the number of parameters of the models. As for models in Tab2, here are the exact parameter sizes of each model:
>
> | Model | CoBIT-Base | CoBIT-Large | CLIP | ALIGN | FILIP | Florence | CoCa-Large | ZeroCap | SimVLM | VLKD | Parti-350M | Parti-750M | LDM (SD) | Coca-2B | Make-A-Scene | Versatile Diffusion | CoDi | DALL-E 2 | CM3Leon-7B | Parti-20B |
> |-------|:----------:|:-----------:|:----:|:-----:|:-----:|:--------:|:----------:|:-------:|:------:|:----:|:----------:|:----------:|:--------:|:-------:|:------------:|---------------------|------|----------|------------|-----------|
> | Size  |    554M    |    1082M    | 428M |  820M |  428M |   893M   |    787M    |   431M  |  632M  | 713M |    350M    |    750M    |   1.4B   |   2.1B  |      4B      | 3.4B                | 4B   | 3.5B     | 7B         | 20B       |
>
> Here are the exact parameter sizes of each model in Tab3:
>
> | Model | CoBIT-Base | CoBIT-Large | CLIP | ALIGN | UNITER | VinVL | CLIP-ViL | ALBEF | BLIP | SimVLM |  OFA | X-LXMERT | Unified-IOXL | CoDi | CoCa-2.1B | BEIT3 | PALI | Parti |
> |-------|:----------:|:-----------:|:----:|:-----:|:------:|:-----:|:--------:|:-----:|:----:|:------:|:----:|:--------:|:------------:|:----:|:---------:|-------|------|-------|
> | Size  |    554M    |    1082M    | 428M |  820M |  303M  |  340M |   136M   |  210M |  417 |  632M  | 930M |   227M   |     2.9B     |  4B  |    2.1B   | 1.9B  | 17B  | 20B   |
>
> In ablations, in default, we use zeroshot ImageNet classification, VQA and zeroshot text-to-image generation as three representative tasks. But when comparing Unicoder-Decoder vs. Encoder-Decoder, finetuning on VQA task updates more parameters in Unicoder-Decoder than Encoder-Decoder, which is not fully fair. To further clarify, we added the zeroshot image captioning task as a fair evaluation for those two designs and have to delete zeroshot ImageNet classification to save space. The performance of zeroshot ImageNet classification is improved by 0.7% when comparing Unicoder-Decoder to Encoder-Decoder. We will add all the details to the camera-ready version.

---

> ### Author Response · Authors · 2023-11-23
> **The reviewer-author discussion will end within 7 HOURS**
>
> Dear Reviewer GAkS,
>
> As the reviewer-author discussion period is nearing its conclusion in under 7 hours, we look forward to your valuable insights and comments on our responses. Please feel free to share your thoughts with us. We sincerely thank you for your dedication and time in reviewing our submission.
>
> Warmest regards,\
> Author(s)

---

### Official Review · Reviewer_wSdD · 2023-10-31

**Soundness:** 4 excellent
**Presentation:** 4 excellent
**Contribution:** 2 fair
**Rating:** 8
**Confidence:** 5

**Summary:**

This paper first propose a multi-modal transformer trained using image-to-text, text-to-image, and contrastive loss, which are three main types of loss function in multimodal area of research. Specially designed architecture allows training the model with three different loss functions all together.

Trained in bidirectional manner, the proposed CoBiT shows great performance on both image-to-text and text-to-image tasks, along with comparable performance on image-text retrieval and other downstream tasks.

**Strengths:**

This paper further advance the research on bidirectional image-text generation task. Shown from previous works, bidirectional image-text training stablize the training while showing comparable performance to other unidirectional models. This paper further add contrastive loss to this bidirectional concept to let the model efficiently learn the latent of multimodal domain.

Trained on large scale dataset (>4B), CoBiT shows great performance on image-to-text and text-to-image generation tasks.

**Weaknesses:**

The problem is that concept of combining two or more different multimodal losses has existed before. The proposed CoBiT architecture seems to be a mixture of L-Verse ( Kim et al. 2022) and CoCa (Yu et al. 2022). While using three different training losses and successfully train a model is a hard work, The experiment section still lacks justification on how it can improve the model's performance.

Since model is trained with more than 4B image-text pairs which is not easily accessible to other researchers (JFT-4B), training CoBiT from scratch with smaller or more general datasets will help readers compare the performance of CoBiT with other works.

**Questions:**

Is there any experimental results of CoBiT model trained with more general and smaller datasets? (CC3M, CC12M, yfcc15m, LAION400M, LAION2B)

---

> ### Author Response · Authors · 2023-11-21
> **Response by Author(s)**
>
> > **W1: Justification on three joint losses.**
>
> **Answer**: Thank you for appreciating the contribution of our work. Our work starts with the motivation of unifying three fundamental objectives and studying their mutual effects, with the hope that those three could benefit each other. We conducted a comprehensive and rigorous ablation in Sec. 4.4, and observed three valuable findings:
>
> * **Finding**: Cross-modal generation objectives can improve image understanding a bit on top of contrastive loss. **Analysis**: Either T2I objective or I2T objective focuses on cross-modal fine-grained semantics, which can explicitly facilitate the global alignment between two modalities.
>
> * **Finding**: Two generation losses, i.e., I2T loss and T2I loss, contradict each other a little bit. **Analysis**: The good signal is that those two don’t contradict that much and still can keep a comparable performance when jointly training. It’s promising because joint training essentially saves half of the parameters compared with using an ensemble of two separate models. On the other hand, those two losses don’t show mutual benefit in performance as we expected. We think the reason is that the granularity of image generation and text generation are different. A single word like cat in the text may correspond to a region in the image that contains hundreds of pixels and tens of image tokens.
>
> * **Finding**: Contrastive loss improves vision-language understanding while it doesn’t influence image generation. **Analysis**: A good global alignment enabled by contrast provides the base for better image-to-text generation. However, image generation requires precise local details beyond just correct semantics, which makes it hard to benefit from global alignment.
>
> As you can see, the mutual effects are kind of mixed, but the parameter efficiency of joint training always holds, i.e., the number of parameters in the ensemble of three specialized models is 2-3 times that of a jointly trained model. We understand that there is plenty of space for better unifying these objectives. Our proposed Unicoder-Decoder is an improvement from the model design. Intuitively, it shares the knowledge between encoding a modality for cross-modality understanding and generating a modality from the other modality, which explicitly bridges both understanding loss and generation loss by sharing the parameter space. Empirically, on all three tasks, it’s all better than the conventional encoder-decoder structure and the average improvement is 6.7% relatively (first row vs. last row of Tab. 5).
>
> > **W2: Smaller or more general datasets.**
>
> **Answer**: Thank you for your valuable feedback. We started with our design hoping to scale the model up to L and even XL sizes, but due to computation limitations and time constraints, we only experimented with Base and Large. From previous literature ([1][2]), larger pre-train models benefit from having larger general-purpose pretraining datasets. Therefore, we were only considering sizable datasets for our experiment purposes. Moreover, LAION datasets had concerns about potential biases embedded in the dataset itself, and hence we did not use it for our training. We will consider running experiments on the smaller datasets with our models in future versions.
>
> [1]. Kaplan, Jared, et al. "Scaling laws for neural language models." arXiv preprint arXiv:2001.08361 (2020). \
> [2]. Chen, Xi, et al. "Pali: A jointly-scaled multilingual language-image model." arXiv preprint arXiv:2209.06794 (2022).

---

> > ### Author Response · Authors · 2023-11-23
> > **The reviewer-author discussion will end within 4 HOURS**
> >
> > Dear Reviewer wSdD,
> >
> > The reviewer-author discussion period is nearing its conclusion in under 4 hours. Please feel free to share your thoughts and any further comments with us. We sincerely thank you for your dedication and time in reviewing our submission.
> >
> > Warmest regards,\
> > Author(s)

---

### Official Review · Reviewer_Ui8A · 2023-10-31

**Soundness:** 4 excellent
**Presentation:** 2 fair
**Contribution:** 3 good
**Rating:** 8
**Confidence:** 4

**Summary:**

This paper introduces a new model known as the Contrastive Bi-directional Image-Text generation model (CoBIT), which is the first to combine three pre-training objectives in one framework: contrastive objectives, image-to-text generative objectives, and text-to-image generative objectives. CoBIT is composed of an image unicoder, a text unicoder, and a cross-modal decoder. The unicoders can perform both encoding and decoding tasks, allowing them to share knowledge and improve both image-to-text and text-to-image generations. The CoBIT model demonstrated superior performance in multiple areas, including image understanding, image-text understanding and text-based content creation, particularly in zero-shot scenarios. The paper concludes that CoBIT's unification of the three objectives has led to strong zero-shot and transferable capacities in unimodal visual understanding, image-text matching, image-text understanding, and text-to-image content creation.

**Strengths:**

1. It is the first to unify three pre-training objectives in one framework, effectively combining contrastive objectives, image-to-text generative objectives, and text-to-image generative objectives. This innovative approach sets it apart from existing models and makes a significant contribution to the field of Vision-and-Language (VL).

2. The performance of CoBIT is outstanding. The authors provide a thorough and detailed explanation of the CoBIT model, including its novel unicoder-decoder structure. They also present extensive experimental results to demonstrate its superior performance in various tasks, including image understanding, image-text understanding, text2image and image2text generation.

3. By allowing the image and text unicoders to switch between encoding and decoding in different tasks, the CoBIT model demonstrates a novel approach to handling both text-to-image and image-to-text generation within a single framework. This not only improves the flexibility of the model but also has the potential to inspire future research in multimodal generation tasks.

**Weaknesses:**

1. Computational Efficiency: The use of Unicoder variants in cross-modal generation scenarios, as mentioned in the appendix of the paper, appears to increase the computational load and the number of parameters used in downstream fine-tuning tasks. However, this aspect is not well elaborated in the main text of the paper. A more thorough discussion on the computational efficiency and trade-offs of using Unicoder compared to Encoder would provide clearer insights into the practicality of the model.

2. Impact of Pre-training Data: The paper shows good ImageNet linear probe performance for CoBIT in Table 3. It would be interesting to see a more detailed analysis of how the pre-training data impacts the model's performance. This could include experiments with different sizes or types of pre-training datasets.

3. Ablation Study: The ablation study in the paper could be strengthened.

4. From Table 1, it can be observed that the proposed base and large sizes of the model have more parameters compared to other multimodal models. This increase in model complexity might have implications for computational resources and efficiency. The authors should provide more explanation or justification for this design choice, and possibly discuss the trade-off between model complexity and performance.

**Questions:**

N/A

---

> ### Author Response · Authors · 2023-11-21
> **Response by Author(s)**
>
> > **W1: Computational Efficiency.**
>
> **Answer**: Thank you for your suggestion. Yes, in training, Unicoder-Decoder and Encoder-Decoder models consume the same number of parameters; in the finetuning of text-to-image and image-to-text tasks, Unicoder-Decoder inherits more parameters than Encoder-Deocder. That is exactly the reason why we want to evaluate zeroshot captioning and zeroshot image generation in the ablation experiments (Tab. 5), which can eliminate the factor brought by updating additional parameters. In those two tasks, Unicoder-Deocoder still outperforms Encoder-Decoder, which validates its effectiveness. We will add this illustration in the camera-ready version of our paper.
>
> > **W2: Impact of Pre-training Data.**
>
> **Answer**: Thank you for this valuable suggestion. Here we listed the ablation on the pre-training datasets. To make a fair comparison, the batch size is kept the same, the training is conducted for 200k steps on base model. As we can see, JFT is beneficial to classification tasks, focusing on basic and precise semantics of image; WebLI has higher-quality image data and is specifically beneficial to text-to-image generation; ALIGN is relatively noisy but covers broad semantics. That’s the reason why we mixed them for a more balanced training.
>
> |                    | ZS. IN. | LP. IN. |  VQA | ZS. T2I (fid, the lower the better) |
> |--------------------|:-------:|:-------:|:----:|:-----------------------------------:|
> | JFT                |   71.6  |   81.4  | 64.8 |                 14.6                |
> | ALIGN              |   70.9  |    81   | 67.2 |                 13.8                |
> | WebLI              |   70.0  |   80.2  | 66.2 |                 13.4                |
>
> > **W3: Ablation Study.**
>
> **Answer**:  We will add the ablation about the loss weights in the appendix and the new ablation about different types of pre-training datasets in the main paper in the camera-ready version.
>
> > **W4: Number of parameters in base/large models.**
>
> **Answer**:  In previous multimodal works, different models have different numbers of parameters for their **base** or **large** models. For example, Coca’s base model has 383M while GIT’s base model has 129M. The reason why our base/large models have a bit more parameters than the previous works’ base or large design is that we have three objectives covering a large number of downstream tasks. Nevertheless, in Tab.2 and Tab.3, we can see that, CoBIT, as a multi-task model, can outperform most of the specialized models within 1B parameters and can even outperform larger models.

---

> ### Comment · Reviewer_Ui8A · 2023-11-22
>
> Thank you for your responses to the reviewers' comments. Here are my thoughts on your responses:
>
> 1. On computational efficiency, I want to know a comprehensive analysis of the computational costs, such as the specific training and inference time, space cost, and the corresponding performance of these models and variants. This data would offer potential users a clear understanding of the trade-offs involved, enabling them to make more informed decisions about the model's utility and application.
>
> 2. On the impact of pre-training data, I also find it intriguing. From Tables 4 and 5 in your paper, it appears that mixing the datasets improved the ZS.T2I performance, while the performance on other metrics after mixing lies between the performances on individual datasets. If your paper is accepted, I would suggest including this analysis in the final version and comparing it with the performance using a mixed pre-training dataset.
>
> 3. I reiterate the opinion that the ablation study is weak and lacks innovative insights. Additionally, given your model's architecture, I am curious as to why you deliberately expanded the number of encoding layers in the decoder to 18, which is typically consistent with the number of layers in the encoder.
>
> 4. On the number of parameters in the base/large models, you mentioned UniLM in your rebuttal to other reviewers. However, as far as I know, UniLM does not introduce additional parameters due to multi-task pre-training.
>
> I greatly appreciate the authors for their responses. I have raised the rating to 6.

---

> > ### Author Response · Authors · 2023-11-23
> > **Thank you for your feedback!**
> >
> > Dear Reviewer Ui8A,
> >
> > We really appreciate your brilliant feedback as well as your effort and time in helping us to improve the quality and clarity of our paper. Regarding your new feedback, please allow us to further explain:
> >
> > > **Question1: A comprehensive analysis of the computational costs:**
> >
> > Here we provided a comprehensive table about the number of parameters, training time, and performance of the main model variants in our paper. We will further verify the inference speed and memory cost precisely and consider adding them in the final version.
> >
> > |                                                  | # Parameters | Training Time (200k steps, shrunken bz) | ZS ImageNet Classification. | VQA      | ZS Caption | ZS Image Generation. (fid, the lower the better) |
> > |--------------------------------------------------|--------------|-----------------------------------------|-----------------------------|----------|------------|--------------------------------------------------|
> > | Contrastive Loss Only                            | 229M         | 70 TPUv4 Days                           | 70.8                        |          |            |                                                  |
> > | Image-to-Text Loss Only                          | 526M         | 91 TPUv4 Days                           |                             | 68       | -          |                                                  |
> > | Text-to-Image Loss Only                          | 418M         | 96 TPUv4 Days                           |                             |          |            | 12.6                                             |
> > | Ensemble of Three Separate Models                | 1173M        | 257 TPUv4 Days                          | 70.8                        | 68       | -          | 12.6                                             |
> > | **CoBIT (Joint training w/ Three losses)**       | **554M**     | **139 TPUv4 Days**                      | **71.1**                    | **66.9** | **37.9**   | **13.3**                                         |
> > | Encoder-Decoder (Joint training w/ Three losses) | 554M         | 135 TPUv4 Days                          | 70.4                        | 65.9     | 32.9       | 13.8                                             |
> >
> > > **Question2:  Impact of pre-training data:**
> >
> > Thank you so much for pointing it out. There might be multiple reasons for this. One potential hypothesis is that the T2I task requires diverse semantics and might be more data-hungry, thus having more non-repeated data samples covering a wider range of semantics might offset its data quality issue (in ALIGN/JFT) and provide more generalizability when mixed with the high-quality data (WebLI). We will add more analysis and discussion about it in the camera-ready version if accepted.
> >
> > > **Question3: Ablation study is weak:**
> >
> > Thank you for your feedback. So far, we presented the ablations on **Mutual Effect of Three Objectives**, **Unicoder-Decoder vs. Encoder-Decoder**, **Initialization**, **Co-efficients of Losses**, **Pre-train Data**. If there is any specific ablation study you think can strengthen the paper, we are happy to add it in the later version.
> >
> > > **Question4: Different numbers of layers in Decoder and Encoder:**
> >
> > That’s a great takeaway. We make the decoder deeper than the encoder because T2I task usually requires more capability on the decoding side in order to generate fine-grained visual details, which is found and utilized in previous T2I works (Parti, etc).
> >
> > > **Question5: Number of parameters in the base/large models:**
> >
> > UniLMs only have one modality as input and output, and the tasks are all NLP-related. However, in CoBIT, two modalities are considered in both input and output, enabling CoBIT to tackle more tasks across modalities, which naturally requires more capabilities and parameters, especially T2I task requires more parameters to generate detailed images (we can find that the SOTA T2I models in Tab2,3 have more parameters on average).

---

### Official Review · Reviewer_PXu6 · 2023-11-02

**Soundness:** 2 fair
**Presentation:** 3 good
**Contribution:** 3 good
**Rating:** 6
**Confidence:** 5

**Summary:**

This paper introduces a complex training framework that combines contrastive learning, image-to-text, text-to-image, and image-to-image approaches. In addition to the intricate model, they also establish a large-scale database. Through the use of these complex models and extensive data, the trained model performs well on various test tasks.

**Strengths:**

- The primary contribution of this paper lies in the integration of several models and the creation of a database tailored for training such models. Building upon this foundation, they conducted parameter optimization to fine-tune the loss, ultimately yielding favorable results.
- This paper is easy to read, with coherent writing throughout. The experimental section offers a rich set of comparisons, although there is room for improvement in the analysis and conclusions.

**Weaknesses:**

While this paper primarily focuses on pre-training and achieves promising results through additional data collection, it still faces the following issues:

- The most significant concern regarding model design in this paper is the limited novelty of the model's contribution. Each module of the model presented in Fig. 2 is already established, and the authors simply combined and fused their losses for joint training. While the training results appear favorable on the extensive data they collected, the paper's drawback remains the lack of model novelty.

- Data is also a concern here. While it's reasonable to leverage more data, it introduces an element of unfairness because the data for the compared models is not consistent. This makes it challenging to draw valid conclusions, and I believe this aspect may also impact the overall findings.

- The optimization process by the authors appears rather intricate, involving the design of numerous loss functions and hyperparameters. This complexity in model design makes replication challenging because the network and parameters are closely tied to the specific dataset they chose. I believe this is also a weakness of the study.

- In the Model Initialization section, the authors mention using CoCa but do not provide a specific explanation for this choice. They do not clarify why they didn't opt for pre-trained BERT models, sentence transformers, or CLIP-based sentence encoders. It would be beneficial if the authors could offer more insight into the rationale behind their selection of COCA as the initialization method and why they didn't consider other pre-trained models for this purpose.

- In Fig. 2, the fact that ViT-VGGAN is frozen to some extent can be considered a weakness, as the existing CoBIT framework assumes the availability of a well-pretrained ViT-VQGAN. This assumption is partially valid, but if the domain changes, it implies that the entire model would need to be re-pretrained. Moreover, if there's a new VQGAN model, the entire model would also require retraining. I am curious about how the authors address this issue, as this paper primarily focuses on pre-training and should ideally rely on existing model checkpoints as much as possible.

- The results of the Linear Probing experiment show a significant improvement, yet the authors have not provided a more substantial explanation. If Linear Probing is the reason for the 1% improvement in performance, how would fine-tuning more parameters affect the outcome?

- The authors trained image captioning using only cross-entropy loss and overlooked RL-based rewards, which are an important component. The authors should have at least discussed how to integrate RL-based methods into this model because RL-based methods, such as self-critical sequence training (SCST), also impose specific requirements on model design.

- Furthermore, when reporting captioning results, the authors should present them on the MSCOCO online test set, as it provides a fair evaluation benchmark, rather than reporting them on a local set.

**Questions:**

Most of the questions that I would like the authors to address have been raised in the "Weaknesses" section.

---

> ### Author Response · Authors · 2023-11-21
> **Response by Author(s) - 1/2**
>
> > **W1: Limited novelty of the model's contribution.**
>
> **Answer**: Thank you for your comments. We appreciate the opportunity to clarify the unique aspects and contributions of our work.
> To our best knowledge, our work is the first exploration to unify all three fundamental objectives jointly. We believe the model design needs to fit the optimization objectives. Although the idea of sharing weights for encoding and decoding has been developed by the previous NLP works (UniLM, etc), they aimed to learn a good and general text representation. Regarding our scenario of optimizing those three cross-modal losses, the proposed Unicoder-Decoder has its unique novelty in **learning beneficial cross-modal knowledge for a joint understanding and generation of two modalities**, as well as **saving parameters**. Compared with the conventional encoder-decoder structure, Unicode-Decoder also obtains better empirical results.
>
> > **W2: The concern about data.**
>
> **Answer**: Thank you for your suggestion. Comparing with other methods under the same training data is not realistic because many works use their own private datasets, such as CLIP, Florence, DALLE, etc.
>
> Since jointly learning cross-modal contrastive optimization, text-to-image generation, and image-to-text generation is a new setting, ours serves as an initial exploration of how those three losses impact each other, and which model structure is more effective and efficient. On those two core problems, all our experiments and ablations are **conducted fairly with the same amount of data**. So the findings of our paper should be valid and useful to the community.
>
> > **W3: Complexity of hyperparameters in model design.**
>
> **Answer**: Thanks for pointing out. It’s common that multi-task learning and multi-objective optimization usually involve more hyperparameters due to multiple losses. We have listed all the hyperparameters in Sec. 4.1 and Tab. 7 of the original submission for a clear reference and also ablated the loss coefficients in Sec. 6.2 (Tab.8).
>
> > **W4: Model Initialization.**
>
> **Answer**: It’s common in previous works that a pre-trained text encoder is used for image generation(DALLE, Parti, SD, etc). We expect that the performance of CLIP and COCA should be similar because they both encode text in a cross-modality way. Due to computation resources and limited time, we didn’t further explore using CLIP to replace Coca. But more essentially, **we did ablate initializing from coca vs training from scratch in Tab.6 and found the training from scratch can bring comparable performance.**
>
> > **W5: Frozen ViT-VQGAN.**
>
> **Answer**: Good point. Freezing VQGAN/VQVAE in image generation model is a common operation not only in auto-regressive methods (DALL-E, Parti), but also in diffusion models (SD). In Ernie-ViLG [1], they try to unfreeze the VQGAN, and in large-scale training, it causes instability in training and the model can’t converge.
>
> [1] Zhang, Han, et al. "Ernie-vilg: Unified generative pre-training for bidirectional vision-language generation." arXiv preprint arXiv:2112.15283 (2021).
>
> > **W6: Explanation about Linear Probing performance.**
>
> **Answer**: Linear probing can usually improve upon zeroshot in general not only in our experiment but also in previous works (CLIP, ALIGN, etc). We think the reason why linear probing can improve the performance over zero-shot performance is that it learns a single-layer adaptation to target categories from the pre-trained knowledge. As for finetuning more parameters, similar to previous works (CLIP, ALIGN, etc), we expect our model to gradually improve until saturated.
>
> > **W7: RL-based captioning.**
>
> **Answer**: Thank you for your suggestion. SCST [2] is usually used to optimize the image captioning model with CIDEr metrics. When applying SCST to CoBIT, no specific model structure change is needed. To be more specific, CoBIT will conduct two feedforwards: one is sampling and the other is greedy inference. CIDEr score will be computed between each of the generated sentences and GT sentence. Then the REINFROCE algorithm is applied, which regards the greedy estimated sentence’s CIDEr score as the baseline and the sampled sentence as the prediction. Therefore, samples from the model that return a higher reward than the greedy estimated sentence will be encouraged, or increased in probability, while samples that result in a lower reward will be suppressed. For the sake of limited budget for experiments and the simplicity of our framework (since our CoBIT already has multiple objectives), we didn’t incorporate this in our experiment but we will consider adding it in the final version.
>
> [2]. Rennie, Steven J., et al. "Self-critical sequence training for image captioning." Proceedings of the IEEE conference on computer vision and pattern recognition. 2017.

---

> ### Author Response · Authors · 2023-11-21
> **Response by Author(s) - 2/2**
>
> > **W8:MSCOCO online test evaluation.**
>
> **Answer**: The reason we report MSCOCO captioning result in Karpathy-split test set is that it’s commonly used in previous works(CoCa, BEIT, PALI, SimVLM, OFA, BLIP1/2, etc) for a more comprehensive comparison. We fairly compare with previous works by making sure no test data leakage in training. The reason for not presenting in MSCOCO online test set is that only very few previous works reported their result on this set (in recent works about image-text foundation models, we only find BUTD(2017), VinVL(2021) and GIT(2022) have official results reported), which makes a comprehensive comparison difficult.

---

> > ### Comment · Reviewer_PXu6 · 2023-11-23
> > **Official Comment by Reviewer PXu6**
> >
> > Thanks for the replies, after reading others' comments and feedback, I have updated my rating.

---

> > > ### Author Response · Authors · 2023-11-23
> > >
> > > We sincerely appreciate your devotion and efforts in reviewing our submission. Your feedback is valuable for improving our work. Thank you again for your time!

---

### Author Response · Authors · 2023-11-22
**Global Response**

We thank all the reviewers for their thoughtful feedback. We are encouraged that they found our work **innovative** (Reviewer Ui8A), **of contribution to the field** (Reviewer Ui8A, Reviewer wSdD), **unified** (Reviewer Ui8A, Reviewer vaSs), **flexible** (Reviewer Ui8A, Reviewer vaSs), **effective with favorable/outstanding performance** (Reviewer PXu6, Reviewer Ui8, Reviewer wSdD, Reviewer GAkS, Reviewer vaSs), **efficient** (Reviewer wSdD, Reviewer GAkS, Reviewer vaSs), **well-written** (Reviewer PXu6, Reviewer GAkS).
We noticed one primary concern of our work about the novelty/advantages of unifying all three objectives and how is it related to our proposed Unicoder-Decoder. We would like to give a common response here to the above question. Besides, we posted our answers to the specific questions in each review respectively, and will incorporate all the feedback in the final version.

> **Question: The novelty/advantages of unifying all three objectives and how is it related to our proposed Unicoder-Decoder?**

**Answer**: Our work starts with the motivation of unifying three fundamental objectives and studying their mutual effects. To our best knowledge, our work is the **first exploration** of this new setup, and we hope to provide **insightful findings for future research** in this domain. We conducted a comprehensive and rigorous ablation in Sec. 4.4, and observed three valuable findings. Before getting into the details, we want to claim that (1). **the parameter efficiency of joint training always holds**, i.e., the number of parameters in the ensemble of three specialized models is 2-3 times that of a jointly trained model; (2) **we don’t intend to make a point that those three objectives must benefit each other**, but rather we want to unravel the relationship between them to shed light on future research.

Here are the three findings and corresponding analysis (also in Sec. 4.4):

* **Finding1**: Cross-modal generation objectives can improve image understanding a bit on top of contrastive loss. **Analysis**: Either T2I objective or I2T objective focuses on cross-modal fine-grained semantics, which can implicitly facilitate the global alignment between two modalities.

* **Finding2**: Two generation losses, i.e., I2T loss and T2I loss, contradict each other a little bit. **Analysis**: The good sign is that those two don’t contradict that much and still can keep a comparable performance when jointly training. It’s promising because joint training essentially saves half of the parameters compared with using an ensemble of two separate models. On the other hand, those two losses don’t show mutual benefit in performance as we expected. We think the reason is that the granularity of image generation and text generation are different. A single word like cat in the text may correspond to a region in the image that contains hundreds of pixels and tens of image tokens.

* **Finding3**: Contrastive loss improves vision-language understanding while it doesn’t influence image generation. **Analysis**: A good global alignment enabled by contrast provides the base for better image-to-text generation. However, image generation requires precise local details beyond just correct semantics, which makes it hard to benefit from global alignment.

As you can see, the mutual effects are kind of mixed in terms of performance, but overall those three losses are compatible in the unified framework. We believe that there is plenty of space for better unifying these objectives.

Our proposed Unicoder-Decoder is an improvement option from the model design. Intuitively, it shares the knowledge between encoding a modality for cross-modality understanding and generating a modality from the other modality, which explicitly bridges both understanding loss and generation loss by sharing the parameter space. Empirically, on all three tasks, it’s all better than the conventional encoder-decoder structure and the average improvement is 6.7% relatively (first row vs. last row of Tab. 5).

---

### Meta-Review · Area_Chair_pMae · 2023-12-01

**Metareview:**

**Summary:** This paper introduces a versatile training model named Contrastive Bi-directional Image-Text generation model (CoBIT).  It unifies three pre-training objectives: cross-modal contrastive learning, image-to-text generation, and text-to-image generation into one framework.  The CoBIT model incorporates an image unicoder, a text unicoder, and a cross-modal decoder that can interchangeably encode and decode information, allowing shared knowledge and enhancing image-text generation tasks.  The model outperforms other models in fields like image understanding, image-text understanding, and content creation, especially in zero-shot scenarios.  The model's excellent performance has been validated through tests against other existing models.

Despite the issues highlighted by reviewers, the strengths and innovations presented in the paper are considered to outweigh the identified weaknesses. Two reviewers offer strong endorsements, two provide marginal support, and one marginally leans towards rejection. The authors have responded to the concerns raised, and although not all issues have been resolved, there is sufficient positive feedback to recommend the paper's acceptance.

**Strengths:** The model is innovative and unifies three pre-training objectives: contrastive learning, image-to-text, and text-to-image generation, within a single framework.  Leveraging a large-scale database, CoBIT uses a flexible unicoder-decoder architecture, leading to excellent performance in various tasks, particularly in zero-shot scenarios.  It showcases parameter efficiency and could significantly impact multimodal research.  The model outperforms its counterparts and the paper is well-structured, offering comprehensive results while suggesting potential for more detailed analysis.

**Weaknesses:**  The paper lacks a theoretical basis for combining image-to-text and text-to-image tasks, an oversight which raises concerns about a potential conflict of objectives with no proposed solutions.  The evaluations seem skewed, favoring strengths and downplaying weaknesses.  Benefits in comparison with other models remain unclear, with only minor gains observed from integration of objectives and using a shared Transformer.  The increased computational burden was left unaddressed, and no analysis is given for the effects of different pre-training datasets.  Despite substantial effort in merging two generative tasks under one model, the paper falls short in offering an exhaustive evaluation, elucidating its unique contributions, and providing a fair comparison against existing models.

**Justification For Why Not Higher Score:**

The paper presents several weaknesses identified by reviewers, and despite the authors' responses, there remain concerns about its final form. Therefore, I cannot recommend it for a higher tier.

**Justification For Why Not Lower Score:**

The paper boasts several strong points facilitating its acceptance, and the authors have responded effectively to most of the reviewers' comments. Even though the reviewers did not engage directly with the authors, there appear to be compelling reasons for accepting the paper.

---

### Decision · Program_Chairs · 2024-01-16

Accept (poster)